# Hydrogen-like Plasmas under Endohedral Cavity

Saptarshi Chowdhury [ID], Neetik Mukherjee [ID] and Amlan K. Roy *

Department of Chemical Sciences, Indian Institute of Science Education and Research (IISER) Kolkata, Mohanpur 741246, India; neetik.mukherjee@iiserkol.ac.in (N.M.)
* Correspondence: akroy@iiserkol.ac.in or akroy6k@gmail.com

**Abstract:** Over the past few decades, *confined quantum systems* have emerged to be a subject of considerable importance in physical, chemical and biological sciences. Under such stressed conditions, they display many fascinating and notable physical and chemical properties. Here we address this situation by using two plasma models, namely a weakly coupled plasma environment mimicked by a Debye-Hückel potential (DHP) and an exponential cosine screened Coulomb potential (ECSCP). On the other hand, the endohedral confinement is achieved via a Woods-Saxon (WS) potential. The *critical screening constant*, dipole oscillator strength (OS) and polarizability are investigated for an arbitrary state. A Shannon entropy-based strategy has been invoked to study the phase transition here. An increase in $Z$ leads to larger critical screening. Moreover, a detailed investigation reveals that there exists at least one bound state in such plasmas. Pilot calculations are conducted for some low-lying states ($\ell = 1 - 5$) using a generalized pseudo spectral scheme, providing optimal, non-uniform radial discretization.

**Keywords:** plasma environment; endohedral confinement; virial-like theorem; oscillator strength; polarizabilities; Shannon entropy

## 1. Introduction

Since its inception, confined quantum systems have emerged as a subject of topical interest. Under such stressed conditions the rearrangement in atomic orbitals leads to significant changes in their energy spectra, causing fascinating behavior in hyper-fine splitting, ionization potential, static and dynamic polarizability, magnetic screening constants etc. [1–5]. Interestingly, in this environment the inert chemical systems like $N_2$, He, and Ar become reactive; also, the sintering effect in organo-metallic catalysts gets reduced. It is needless to mention that the discovery of such artificial systems has opened up several interesting possibilities in almost all fields of science, and in technology. For example, in the realm of quantum computers, researchers have proposed an alternative way where the qubits are formed using the concept of confinement. This is possible because of the isolating property of endohedral fullerenes which almost perfectly isolate the atomic properties, particularly the spin of an atom/ion trapped inside, as it plays a vital role in carrying the information [6–9]. More applications can be found in photo-voltaic materials [10], semiconducting nano-crystals [11], fusions experiments [12], hydrogen storage [13] and medicinal science [14].

Ever since the early stage of their discovery, endohedral systems like fullerene [15] and supra-molecular structures like crown ether, graphene tubes were utilized to trap quantum systems. The variety in size and shape of cavity (which is directly dependent on number of carbon atoms) has helped researchers to trap systems as small as H atom to larger ones like $H_2O$, alkali atoms, Xe [16–18] etc. Several spectroscopic quantities like oscillator strength (OS) [19,20], polarizability [21], photo ionization cross-section [22] and stopping cross-sections [23] were reported in such scenarios. Apart from that, Shannon entropy and Fisher information [24] were also studied. Interestingly, these systems can also explain the shell-confinement model. In recent works, Gaussian well and Woods-Saxon

(WS) model potentials are used to mimic endohedral confinement, besides the square well potential [24–28]. It appears that, generally speaking, the WS model is a better fit when compared with experimental observations in such confinement studies. The presence of a smoothing parameter in the potential gives an advantage in changing the nature of the curve from soft to sharp and *vice-versa*.

In a plasma environment, Coulomb and thermal energy play a pivoting role in determining the strength of coupling. Moreover, factors like electron density ($n$) and temperature ($T$) are critical because they equilibrate the bound states of a given system. The coupling constant is expressed as the ratio of average Coulomb energy and average thermal energy,

$$\Gamma = \frac{E_{Coulomb}}{E_{thermal}} = \frac{Q^2}{4\pi\epsilon_0 a k_b T_e}, \tag{1}$$

where $Q$ signifies the charge of particle, $a$ refers to the inner-particle separation, $K_b$ is the usual Boltzmann constant. This presents two different scenarios: (i) $\Gamma < 1$, when $n$ is low and $T$ is high, making thermal energy larger than Coulomb energy; this is generally known as weak coupled plasma, (ii) $\Gamma > 1$, when $n$ is high but $T$ is low, causing thermal energy smaller than Coulomb energy; this is usually referred to as strongly coupled plasma.

A prototypical example of weakly coupled plasma is Debye-Hückel potential (DHP), provided by introducing an exponential term in the Coulomb potential [29]. It has been investigated extensively; the literature is quite vast. Energy spectra, [30–33], transition probabilities with electron-impact excitation [34,35], OS, polarizability [36–41], inelastic electron-ion scattering [42,43], two-photon transition [44,45], Shannon entropy and Fisher information [46], mean-excitations energy [47] were studied with great interest. Research was also conducted to estimate the critical screening parameter [48], $\lambda^{(c)}$, beyond which no bound state exists.

When the Debye radius becomes comparable to the *de broglie* wavelength, a quantum effect starts to appear. Thus, we use exponential cosine screened Coulomb potential (ECSCP), where a cosine function is multiplied with the usual DHP; this invokes the composite screening and wake effect around a slow-moving charge in a low-density quantum plasma. This is due to the fact that statistical pressure is preponderated by the quantum force of plasma electrons. This potential is examined intensely using the perturbation and variational method, the shooting method, the super-symmetric perturbation method, the Padé scheme, basis-expansion with Slater-type orbitals, the generalized pseudo-spectral (GPS) method, the Laguerre polynomial and so on. Besides eigenvalues and eigenfunctions, several important spectroscopic properties like OS, polarizability and the photo-ionization cross-section were under examination [48–58].

In a recent work, astrophysicists have discovered the existence of fullerene entrapped plasmas in interstellar space. This also includes radioactive atoms and fusion products [59–61]. This has triggered the investigation of such systems in both free and confined conditions. As a consequence, properties like $\lambda^{(c)}$, OS (dipole), photo-ionization cross-section, static dipole polarizability, mean excitation energy and electronic stopping cross-section were investigated for different well depths, including the avoided crossing region in endohedral H atom embedded in a Debye plasma [62]. To the best of our knowledge, the literature is rather scarce for the current system under study. In this context, we want to explore the trapping of an H-like plasma (modeled through DHP and ECSCP) under endohedral environment. In order to proceed, at first we engage the accurate and efficient GPS method for calculating eigenvalues and eigenfunctions of the corresponding Schrödinger equation (SE). Then quantities like $\lambda^{(c)}$, OS and polarizability are studied. Later we also explore the behavior of such a system under external high pressure. For this, we have used two different models, namely, (i) DHP under WS (DHPWS) and (ii) ECSCP under WS (ECSCPWS). Following [63], where the $\lambda^{(c)}$s were estimated through a Shannon entropy criterion for *free* DHP and ECSCP, here we employ the same procedure to report these under a *confinement* environment. This is performed for $\ell$ values ranging from 0–4, while $Z$ remains 1 or 2. The OS and polarizability are calculated varying $\lambda$ for some selective

values of confinement radius, $r_c = 7.7, 8$, and for some large $r_c = 5 \times 10^5 \, (\approx \infty)$ with $Z = 1, 2$ for $1s$, $2s$ and $2p$ states. The article is organized as follows: Section 2 gives a brief introduction to the theoretical formalism, Section 3 discusses the obtained results, and finally some concluding remarks are made in Section 4.

## 2. Theoretical Formalism

The time-independent radial SE for the spherically confined plasma system is expressed as (in a.u.):

$$\left[ -\frac{1}{2} \frac{d^2}{dr^2} + \frac{\ell(\ell+1)}{2r^2} + V_p(r) + V_c(r) \right] u_{n,\ell}(r) = \mathcal{E}_{n,\ell} \, u_{n,\ell}(r). \tag{2}$$

Here, the total wave function is $\Psi(r) = R_{nl}(r)Y_{ml}(\theta, \phi)$ of which $R_{nl}(r) = u_{nl}(r)/r$ is the radial wave function and $Y_{ml}(\theta, \phi)$ is the spherical harmonics. The first term of this equation denotes the usual kinetic energy of electrons. To represent the excited states, we need to add the angular contribution and thus the second term. The last two terms signify the potential models characterizing the plasma through $V_p(r)$, and the fullerene cavity through $V_c(r)$. To calculate energy and spectroscopic properties, we have employed a non-uniform, optimal radial discretization scheme via GPS method. Its accuracy and efficiency in calculating various bound-state properties have been reported in our previous papers in the case of several central potentials in both *free* and *confined* condition. For more detailed information, readers are referred ([64–69] and therein).

In this article, we employ three different plasma model potentials to simulate the experimental environment. First, a DHP model, where the collective screening by plasma electron is mapped and expressed as,

$$V_{dh}(r) = \begin{cases} = -\frac{Z}{r} e^{-\lambda r}, & r \leq r_c \\ = 0, & r > r_c. \end{cases} \tag{3}$$

Here, the probability of finding plasma electrons inside the Debye sphere is negligible. Moreover, the inequalities ensure that the charge cloud remains confined. The second model used is called ECSCP, which accounts for the fact that with an increase in plasma density the Debye radius becomes commensurable to *de Broglie* wavelength, which emerges in the form of quantum effects [70]. This is accomplished by taking a potential of the following form,

$$V_{ec}(r) = \begin{cases} = -\frac{Z}{r} e^{-\lambda r} \cos(\lambda r), & r \leq r_c \\ = 0, & r > r_c, \end{cases} \tag{4}$$

The cosine function helps to achieve a stronger screening effect than the DHP. Then, we also make use of the (WS) potential to model fullerene as an external confinement. Within this framework, to get the desired effect of endohedral fullerene confinement, parameters like the radius of the fullerene cavity $R_c$, thickness of fullerene shell $\Delta$, as well as two fitting parameters, namely $V_0$ (defining well depth) and $\gamma$ (a suitable smoothing parameter) are necessary. It is mathematically expressed as,

$$V_c(r) = \frac{V_0}{1 + e^{-[r-(R_c+\Delta)]/\gamma}} - \frac{V_0}{1 + e^{-(r-R_c)/\gamma}} \tag{5}$$

In this work, parameters are fixed as follows: $V_0 = 0.302$, $R_c = 5.8$, $\Delta = 1.89$ and $\gamma = 0.1$, which are used to describe endohedral confinement in numerous references [71–76].

### 2.1. Oscillator Strength and Polarizability

The static multi-pole polarizability can be expressed in following form,

$$\alpha_i^{(k)} = \alpha_i^{(k)}(\text{bound}) + \alpha_i^k(\text{continuum}). \tag{6}$$

There are mainly two approaches to calculating static polarizability. The first one is direct computation using a perturbation theory framework. The second one is a sum-over-state method. The latter is more convenient to express and has a compact form in nature. That is why we have used this approach. Accordingly, the $2^k$-pole static polarizability is expressed as,

$$\alpha_i^{(k)} = \sum_f \frac{f_{fi}^{(k)}}{(\mathcal{E}_f - \mathcal{E}_i)^2} - c \int \frac{|\langle R_i | r^k Y_{kq}(\mathbf{r}) | R_{\epsilon n} \rangle|^2}{(\mathcal{E}_{\epsilon n} - \mathcal{E}_i)} \, d\epsilon \tag{7}$$

On the right, the two terms signify the polarizability contributions from bound states and continuum states, respectively. Moreover, $f_{fi}^{(k)}$ denotes multi-pole OS where $k$ is a positive integer, while $\Delta E_{fi} = \mathcal{E}_f - \mathcal{E}_i$ is the energy difference between the transition states.

The $2^k$-pole OS can be expressed as:

$$f_{ni}^{(k)} = \frac{8\pi}{(2k+1)} \Delta\mathcal{E}_{fi} |\langle \psi_f | r^k Y_{kq}(\mathbf{r}) | \psi_i \rangle|^2 \tag{8}$$

Designating the initial and final states as $|n\ell m\rangle$ and $|n'\ell'm'\rangle$, one derives the mean OS as,

$$\bar{f}_{fi}^{(k)} = \frac{8\pi}{(2k+1)} \Delta\mathcal{E}_{fi} \frac{1}{2\ell+1} \sum_m \sum_{m'} |\langle n'\ell'm' | r^k Y_{kq}(\mathbf{r}) | n\ell m \rangle|^2. \tag{9}$$

This is a necessary part because the mean OS does not depend on the magnetic quantum number. Thereafter we apply Wigner-Eckart theorem and sum rule for *3j* symbol [77] leading us to,

$$\bar{f}_{fi}^{(k)} = 2 \frac{(2\ell'+1)}{(2k+1)} \Delta\mathcal{E}_{fi} |\langle r^k \rangle_{n\ell}^{n'\ell'}|^2 \left\{ \begin{array}{ccc} \ell' & k & \ell \\ 0 & 0 & 0 \end{array} \right\}^2. \tag{10}$$

The transition matrix element may be expressed by the following,

$$\langle r^k \rangle = \int_0^\infty R_{n'\ell'}(r) r^k R_{n\ell}(r) r^2 dr. \tag{11}$$

In this article, we aim to compute dipole polarizability and OS for $1s, 2s, 2p$ states. The relevant selection rule for dipole OS ($k = 1$) for these states are ($i = 1$ or 2),

$$\bar{f}_{np-is}^{(1)} = 2 \Delta\mathcal{E}_{np-is} |\langle r \rangle_{is}^{np}|^2 \left\{ \begin{array}{ccc} 1 & 1 & 0 \\ 0 & 0 & 0 \end{array} \right\}^2 = \frac{2}{3} \Delta\mathcal{E}_{np-is} |\langle r \rangle_{is}^{np}|^2. \tag{12}$$

It is important to mention that there exists an OS sum rule [78,79] in literature. To show that all the transition states are rendered properly it is customary to verify what it holds. It is denoted by $S^{(k)}$ and the respective sum rule is given by,

$$S^{(k)} = \sum_n f^{(k)} = k \langle \psi_i | r^{(2k-2)} | \psi_i \rangle, \tag{13}$$

## 2.2. Shannon Entropy

Shannon entropy ($S$) is a statistical quantity and a function of density. It is the arithmetic mean of uncertainty associated with a given state. Therefore, it can be aptly used to measure the change in density at the point of transition from bound to continuum state.

It has been observed before that, at that point, $S_r$ jumps suddenly. It is recently used to investigate confined systems [80–83]. Mathematically, these are written as,

$$
\begin{aligned}
S_r &= -\int_{R^3} \rho(r) \ln \rho(r) r^2 \, dr, \qquad S_p = -\int_{R^3} \Pi(p) \ln \Pi(p) p^2 \, dp, \\
S_{\theta,\phi} &= -\int \chi(\theta) \ln \chi \sin(\theta) \, d\theta, \qquad \chi(\theta) = |\Theta(\theta)|^2, \\
S_t &= S_r + S_p + 2S_{\theta,\phi} \geq 3(1 + \ln \pi).
\end{aligned}
\tag{14}
$$

Here, $\rho(r)$ is the normalized position-space density, while $\Pi(p)$ corresponds to momentum-space density. This work incorporates $S_r$ to characterize $\lambda^{(c)}$ in the mixed potential condition.

## 3. Result and Discussion

The result will be discussed in three subsections. At the onset, it may be noted that all calculations are achieved by keeping the pseudo-spectral parameter fixed at $L = 1$ (mapping parameter) and $N = 300$ (number of grid points). At first, we present the calculated $\lambda^{(c)}$s for both DHPWS and ECSCPWS through Shannon entropy. This will allow us to investigate the phase transition in plasma systems. Then we look into the dipole OS and polarizability for these pair of H-like plasmas embedded in the fullerene cavity, taking $Z = 1, 2$. It is necessary to mention that, for the purpose of calculation, we have kept $r_c > (R_c + \Delta)$. Pilot calculations are conducted in $\ell = 0$–$5$ states using the experimental $V_0, \Delta, R_c, \gamma$ values mentioned in Section 2.

### 3.1. Critical Screening Constant in DHPWS and ECSCPWS

In a plasma system, the number of bound state decreases with an increase in $\lambda$. The characteristic $\lambda$ at which a bound state vanishes is termed as $\lambda^{(c)}$, i.e., beyond which no bound state exists. In the present endeavor, our objective is to find the effect of the fullerene environment on $\lambda^{(c)}$. In practice, $\lambda^{(c)}$ is determined by the sign change argument of energy ($-$ve for bound state and $+$ve continuum state). Interestingly, in a recent paper [63], the authors have successfully used $S_r$ to determine the same. At $\lambda^{(c)}$, it jumps suddenly to a higher value indicating the conversion from a bound to a continuum state. Thus, at that point $\left(\frac{dS_r}{d\lambda}\right) \to \infty$, explaining a first-order phase transition. In general, this occurs when $\left(\frac{dS_r}{dT}\right) \to \infty$. We know that, $\lambda$ is a function of $T$. Therefore, one can identify $\lambda^{(c)}$ as a first-order phase transition point. Figure 1 depicts the behavior of calculated $S_r$ against $\lambda$ for circular node-less states, having $\ell = 0$–$2$ in DHPWS (panel I) and ECSCPWS (panel II), having $Z = 1, 2$. Similarly, Figure 2 portrays the variation of $S_r$ as functions of $\lambda$ for some other states like, $2s, 3s, 3p, 4d, 4f, 5f, 5g, 6g$, with $Z = 1$ and 2 in DHPWS (panels I) and ECSCPWS (panels II) systems. A detailed analysis of these figures reveals the following:

1.  At the onset, it should be mentioned that the qualitative behaviour of $S_r$ with $\lambda$ in DHPWS and ECSCPWS are quite similar.
2.  Panels (I), (II) in Figure 1 show that there exists at least three bound states in either of the fullerene trapped plasmas. Because, in both cases, circular or node-less states with $\ell = 0$–$2$ are never going to be deleted. As a consequence, no abrupt jump in $S_r$ is observed. In these states $S_r$ increases with $\lambda$ and finally converges to the respective limiting values.
3.  Panels (I), (II) of Figure 2 suggest that, for a given state, there exists a characteristic $\lambda$ at which the $S_r$ value jumps suddenly, signifying the phase transition. The position of these $\lambda^{(c)}$ gets right shifted with a rise in $Z$. Here, a first order phase transition happens in both the plasmas involving $2s, 3s, 3p, 4d, 4f, 5f, 5g, 6g$ states.
4.  These observations lead us to the conjecture that, in these two fullerene trapped plasmas, phase transition occurs for all $\ell \geq 3$ states. However, for $\ell = 0$–$2$ states, a similar phenomenon occurs only when $(n - \ell - 1) \geq 1$.

Apart from the above procedure, we have also estimated $\lambda^{(c)}$ using the usual sign-change argument of energy for the same set of states explored above. Results are collected in Table 1 at two selected $Z$ (1, 2) for both DHPWS and ECSCPWS. It is worth mentioning that these results corroborate all the inferences drawn from Figures 1 and 2. Moreover, the present result of $\lambda^{(c)}$ in DHPWS is in complete consonance with the observation (obtained by energy sign change) given in [62], as evident from the entries provided in the footnote of Table 1. However, the $\lambda^{(c)}$ pattern in ECSCPWS could not be found in the literature; here they are reported for the first time. Now, we move to investigate dipole OS and polarizability for these pair of fullerene-trapped plasmas in both the confined condition and at the free regime as a limiting case. Note that some of the entries are left blank indicating no $\lambda^{(c)}$ in such occasions.

**Table 1.** Calculated $\lambda_{n,\ell}^{(c)}$ for H-like ions for $1s, 2s, 3s, 2p, 3p, 3d, 4d, 4f, 5f, 5g, 6g$ states in DHPWS and ECSCPWS ($L = 1, N = 300$). See text for details.

| | | **DHPWS** | | | | **ECSCPWS** | |
|---|---|---|---|---|---|---|---|
| $Z$ | State | $\lambda_{n,\ell}^{(c)}$ | $\mathcal{E}_{n,\ell}$ | $Z$ | State | $\lambda_{n,\ell}^{(c)}$ | $\mathcal{E}_{n,\ell}$ |
| 1 | $1s$ | — | — | 1 | $1s$ | — | — |
| 2 | | — | — | 2 | | — | — |
| 1 | $2s$ | 0.9111 [a] | −0.000000254 | 1 | $2s$ | 0.6287 | −0.000000038 |
| 2 | | 2.1247 | −0.000000197 | 2 | | 1.3618 | −0.000000165 |
| 1 | $3s$ | 0.1565 [a] | −0.000000439 | 1 | $3s$ | 0.0762 | −0.000000130 |
| 2 | | 0.3900 | −0.000000055 | 2 | | 0.2688 | −0.000000238 |
| 1 | $2p$ | — | — | 1 | $2p$ | — | — |
| 2 | | — | — | 2 | | — | — |
| 1 | $3p$ | 0.1160 [a] | −0.000000096 | 1 | $3p$ | 0.0730 | −0.000001331 |
| 2 | | 0.3575 | −0.000011586 | 2 | | 0.2633 | −0.000053585 |
| 1 | $3d$ | — | — | 1 | $3d$ | — | — |
| 2 | | — | — | 2 | | — | — |
| 1 | $4d$ | 0.0732 | −0.000000291 | 1 | $4d$ | 0.0466 | −0.000003059 |
| 2 | | 0.1197 | −0.000005269 | 2 | | 0.0782 | −0.000064445 |
| 1 | $4f$ | 0.1947 | −0.000000856 | 1 | $4f$ | 0.1267 | −0.000007139 |
| 2 | | 0.2994 | −0.000023163 | 2 | | 0.1681 | −0.000018290 |
| 1 | $5f$ | 0.0472 | −0.000016099 | 1 | $5f$ | 0.0323 | −0.000001631 |
| 2 | | 0.0831 | −0.000007057 | 2 | | 0.0540 | −0.000038551 |
| 1 | $5g$ | 0.0333 | −0.000000778 | 1 | $5g$ | 0.0278 | −0.000006697 |
| 2 | | 0.1231 | −0.000005413 | 2 | | 0.0903 | −0.000078050 |
| 1 | $6g$ | 0.0288 | −0.000005039 | 1 | $6g$ | 0.0210 | −0.000001928 |
| 2 | | 0.0588 | −0.000040341 | 2 | | 0.0403 | −0.000068680 |

[a] Literature result for $Z = 1$: $\lambda_{2s}^{(c)} = 0.909926$, $\lambda_{3s}^{(c)} = 0.1534008$ and $\lambda_{3p}^{(c)} = 0.115595$.

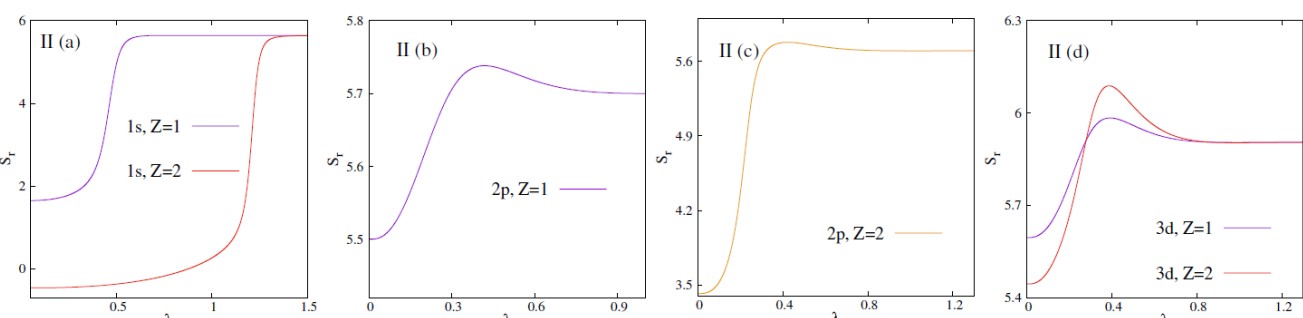

**Figure 1.** *Cont.*

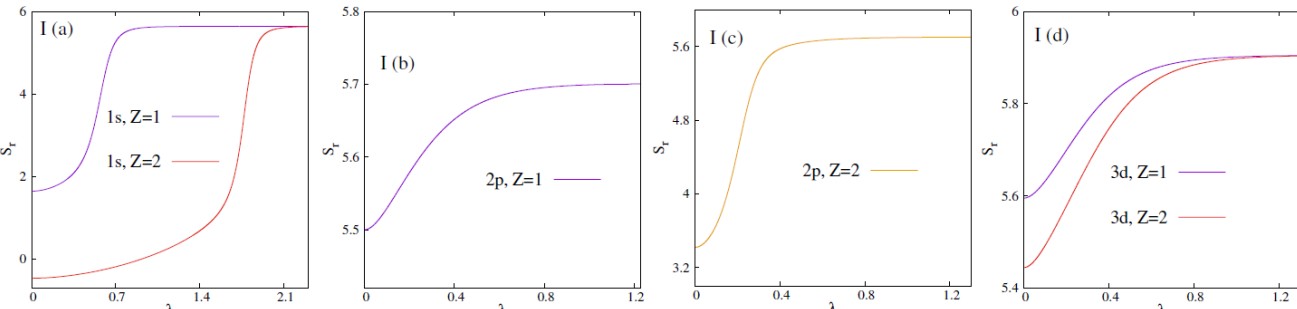

**Figure 1.** Plot of $S_r$ as function of $\lambda$ for $1s, 2p, 3d$ states; bottom (I) and top (II) panels refer to DHPWS and ECSCPWS, having $Z = 1$, 2. See text for details.

**Figure 2.** Plot of $S_r$ as function of $\lambda$ for $2s, 3s, 3p, 4d, 4f, 5f, 5g, 6g$; bottom (I) and top (II) panels refer to DHPWS and ECSCPWS, having $Z = 1$, 2. See text for details.

### 3.2. Dipole Oscillator Strength

In this section, we aim to discuss dipole OS for free and confined fullerene-trapped plasmas. It may be noted that, $f^{(1)}$ acts as a cornerstone in estimating several spectroscopic properties, like polarizability, mean excitation energy etc. The $(+)$ve value of $f^{(1)}$ indicates absorption, whereas a $(-)$ve value signifies emission. With progress in $f^{(1)}$, the probability of radiative transition increases in comparison to its non-radiative counterpart. On the contrary, $f^{(1)} \rightarrow 0$ explains non-radiative transitions. Here, calculations are conducted at three selected $r_c$ values, namely $7.7, 8$ and infinity.

In the case of $\ell = 0$, the selection rule for dipole transition is $\Delta \ell = \pm 1$. Thus, the transition may be possible only to $\ell = 1$ states. However, for p and d states, transitions may occur to $\ell = 0, 2$ and $\ell = 1, 3$ states, respectively. Table 2, at first, reports the calculated $f^{(1)}$ for $1s$, $2p$ and $3d$ states at several selected $\lambda$, keeping $r_c$ fixed at 7.7, 8, $\infty$ and $Z = 1, 2$, addressing all possible modes of transition allowed by the selection rule. Here, columns 2–6 represent DHPWS and columns 7–10 signify ECSCPWS. In both situations, at a given $Z$ and $r_c$, $f^{(1)}_{1s \rightarrow 2p}$ decreases to reach a minimum, and then rises to a maximum before achieving the limiting value. Further, at a fixed $r_c$, on moving from $Z = 1$ to 2, the position of the minimum gets right shifted. On the contrary, $f^{(1)}$ for $2p \rightarrow 1s$ indicates emission. Here, at a fixed $Z$ and $r_c$, $f^{(1)}$ increases to attain a maximum and declines thereafter; the position of the maximum moves toward the right with $Z$. The $2p \rightarrow 3d$ transition suggests absorption. In this case, $f^{(1)}$ imprints a distinctly different pattern in DHPWS and ECSCPWS. In DHPWS a shallow maximum is seen at $Z = 1, r_c = 7.7$. However, at $r_c = 8$ it decreases with $\lambda$. On the contrary, at $Z = 2$ and $r_c = 7.7, 8$ it passes through a maximum. In ECSCPWS for $Z = 1$ (when $r_c = 7.7$, 8) it reduces with $\lambda$. However, at $Z = 2$, it decreases to reach a minimum then climbs a maximum and then again declines to a minimum. As expected, $f^{(1)}_{3d \rightarrow 2p}$ explains emission. In this case, we observe more or less an opposite pattern to $f^{(1)}_{2p \rightarrow 3d}$. In DHPWS, $f^{(1)}_{3d \rightarrow 4f}$ abates with $\lambda$ at $Z = 1, r_c = 7.7, 8$ and $Z = 2, r_c = 8$. However, in case of $Z = 2, r_c = 7.7$ it passes through a maximum. Now, we move to discuss the behavior of $f^{(1)}$ in either of the plasmas at $r_c \rightarrow \infty$ involving these five transitions. Table 3 records the same for these two plasmas at some selected $\lambda$ value, keeping $Z = 1$ and 2. It is observed that variation of $f^{(1)}$ in *free* systems is different from their *confined* counterparts. Moreover, $f^{(1)}_{1s \rightarrow 2p}$ at $Z = 1$ drops sharply to reach a minimum and then climbs up to a maximum. Similarly, for $Z = 2$, it passes through a minimum. For the $2p \rightarrow 1s$ transition we observe a reverse trend. Now, for $2p \rightarrow 3d$ transition, in DHPWS (i) we observe a decreasing pattern at $Z = 1$ (ii) but at $Z = 2$, we see a maximum. In ECSCPWS, at $Z = 1$, a minimum is noticed. At $Z = 2$, there appears a maximum followed by a minimum. Here, $f^{(1)}_{3d \rightarrow 2p}$ imprints similar trends to $f^{(1)}_{2p \rightarrow 1s}$. Finally, we derive a decreasing curve for $3d \rightarrow 4f$ transition in both the plasmas.

The above observations of Tables 2 and 3 are rather sketchy. To derive a clearer picture, we have plotted $f^{(1)}$ for DHPWS and ECSCPWS in Figures 3 and 4 as function of $\lambda$, respectively, at three selected $r_c$ values, keeping $Z = 1$ and 2. The transitions shown correspond to (a) $1s \rightarrow 2p$ (b) $2p \rightarrow 1s$ (c,d) $2p \rightarrow 3d$ (e,f) $3d \rightarrow 2p$ and (g,h) $3d \rightarrow 4f$. Panels (c,e,g) correspond to $Z = 1$ while (d,f,h) produce the same for $Z = 2$. It is found that $f^{(1)}$ in DHPWS and ECSCPWS are quite similar. A careful examination reveals that, all these plots corroborate the inferences drawn from Tables 2 and 3.

**Table 2.** $f^{(1)}$ values for DHPWS and ECSCPWS for fixed $r_c = 7.7, 8$ and $Z = 1, 2$, using $L = 1$ and $N = 300$.

| Transition | $\lambda$ | DHPWS | | | | ECSCPWS | | | |
|---|---|---|---|---|---|---|---|---|---|
| | | $r_c = 7.7$ | | $r_c = 8$ | | $r_c = 7.7$ | | $r_c = 8$ | |
| | | $Z = 1$ | $Z = 2$ | $Z = 1$ | $Z = 2$ | $Z = 1$ | $Z = 2$ | $Z = 1$ | $Z = 2$ |
| | 0.01 | 0.472909 | 0.405489 | 0.393008 | 0.399455 | 0.473031 | 0.405670 | 0.393153 | 0.399658 |
| | 0.1 | 0.463893 | 0.389264 | 0.382538 | 0.380980 | 0.466412 | 0.399214 | 0.385322 | 0.392167 |
| $1s \rightarrow 2p$ | 0.5 | 0.506522 | 0.146559 | 0.445168 | 0.104184 | 0.506896 | 0.062774 | 0.475507 | 0.036498 |
| | 1.0 | 0.859970 | 0.072607 | 0.881023 | 0.047763 | 0.941505 | 0.069773 | 0.911234 | 0.051167 |
| | 2.5 | 0.937700 | 0.910538 | 0.902761 | 0.915478 | 0.918853 | 0.924631 | 0.884901 | 0.889980 |
| | 0.01 | −0.157636 | −0.135163 | −0.131002 | −0.133151 | −0.157677 | −0.135223 | −0.131051 | −0.133219 |
| | 0.1 | −0.154631 | −0.129754 | −0.127512 | −0.126993 | −0.155470 | −0.133071 | −0.128440 | −0.130722 |
| $2p \rightarrow 1s$ | 0.5 | −0.168840 | −0.048853 | −0.148389 | −0.034728 | −0.168965 | −0.020924 | −0.158502 | −0.012166 |
| | 1.0 | −0.286656 | −0.024202 | −0.293674 | −0.015921 | −0.313835 | −0.023257 | −0.303744 | −0.017055 |
| | 2.5 | −0.312566 | −0.303512 | −0.300920 | −0.305159 | −0.306284 | −0.308210 | −0.294967 | −0.296660 |
| | 0.01 | 1.088693 | 0.800227 | 1.082666 | 0.732754 | 1.088689 | 0.800197 | 1.082677 | 0.732695 |
| | 0.1 | 1.088778 | 0.806429 | 1.081425 | 0.743057 | 1.088627 | 0.798441 | 1.081926 | 0.731705 |
| $2p \rightarrow 3d$ | 0.5 | 1.071869 | 1.055338 | 1.054450 | 1.060247 | 1.055825 | 1.067797 | 1.036342 | 1.050053 |
| | 1.0 | 1.052710 | 1.063993 | 1.034850 | 1.045177 | 1.039857 | 1.036862 | 1.023702 | 1.020954 |
| | 2.5 | 1.043849 | 1.044600 | 1.027258 | 1.027832 | 1.042933 | 1.042720 | 1.026567 | 1.026410 |
| | 0.01 | −0.653216 | −0.480136 | −0.649599 | −0.439652 | −0.653213 | −0.480118 | −0.649606 | −0.439617 |
| | 0.1 | −0.653267 | −0.483857 | −0.648855 | −0.445834 | −0.653176 | −0.479064 | −0.649156 | −0.439023 |
| $3d \rightarrow 2p$ | 0.5 | −0.643121 | −0.633202 | −0.632670 | −0.636148 | −0.633495 | −0.640678 | −0.621805 | −0.630032 |
| | 1.0 | −0.631626 | −0.638396 | −0.620910 | −0.627106 | −0.623914 | −0.622117 | −0.614221 | −0.612572 |
| | 2.5 | −0.626309 | −0.626760 | −0.616355 | −0.616699 | −0.625760 | −0.625632 | −0.615940 | −0.615846 |
| | 0.01 | 1.361677 | 1.378364 | 1.350361 | 1.374690 | 1.361703 | 1.378352 | 1.350391 | 1.374693 |
| | 0.1 | 1.359552 | 1.378857 | 1.347923 | 1.373824 | 1.360468 | 1.378527 | 1.348914 | 1.374145 |
| $3d \rightarrow 4f$ | 0.5 | 1.342821 | 1.355667 | 1.330534 | 1.342513 | 1.333760 | 1.336163 | 1.321648 | 1.322866 |
| | 1.0 | 1.334518 | 1.337464 | 1.322958 | 1.325371 | 1.329986 | 1.328230 | 1.319261 | 1.317836 |
| | 2.5 | 1.331894 | 1.331979 | 1.320806 | 1.320868 | 1.331802 | 1.331794 | 1.320741 | 1.320736 |

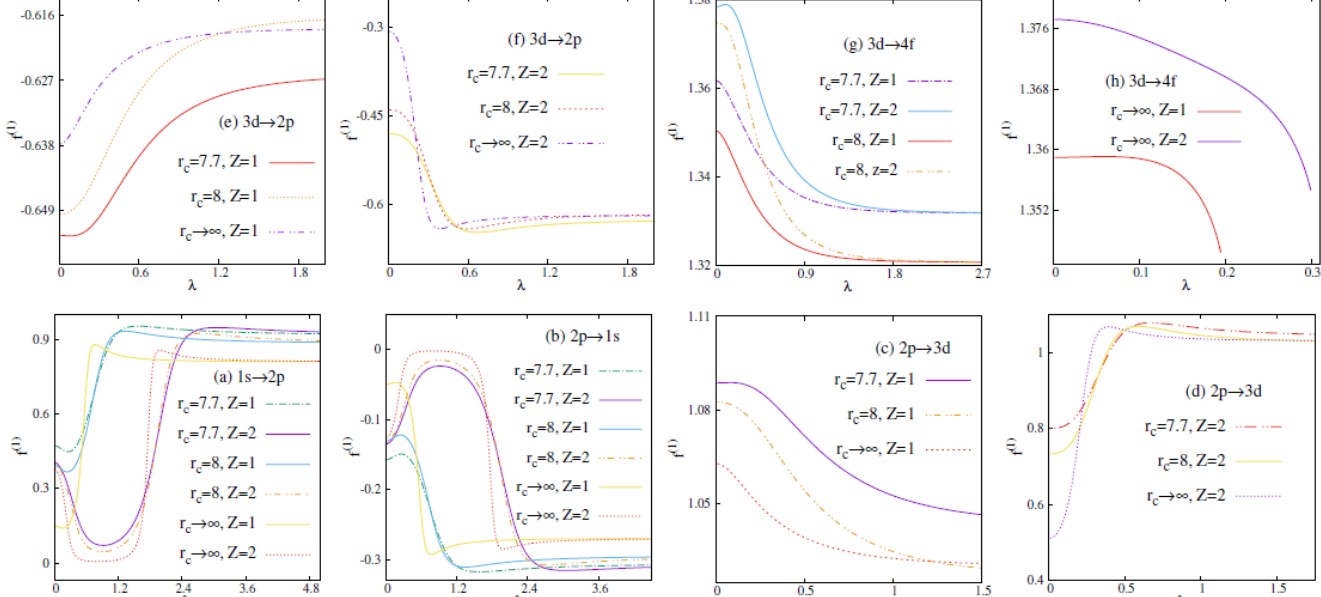

**Figure 3.** Plot of $f^{(1)}$ as function of $\lambda$ in DHPWS potential for selected transitions mentioned in the panels (**a**–**h**), for $Z = 1, 2$. See text for details.

**Table 3.** $f^{(1)}$ values for DHPWS and ECSCPWS for $r_c \to \infty$ and $Z = 1, 2$, using $L = 1$ and $N = 300$.

| Transition | DHPWS | | | | ECSCPWS | | | |
|---|---|---|---|---|---|---|---|---|
| | $\lambda$ | $Z = 1$ | $\lambda$ | $Z = 2$ | $\lambda$ | $Z = 1$ | $\lambda$ | $Z = 2$ |
| $1s \to 2p$ | 0.01 | 0.149983 | 0.01 | 0.377669 | 0.01 | 0.150086 | 0.01 | 0.378002 |
| | 0.1 | 0.143844 | 0.1 | 0.345192 | 0.1 | 0.143408 | 0.1 | 0.363852 |
| | 0.5 | 0.405345 | 0.5 | 0.014097 | 0.5 | 0.795304 | 0.5 | 0.003368 |
| | 1.0 | 0.853185 | 1.0 | 0.008094 | 1.0 | 0.815081 | 1.0 | 0.015977 |
| | 2.5 | 0.813876 | 2.5 | 0.829531 | 2.5 | 0.809188 | 2.5 | 0.810333 |
| $2p \to 1s$ | 0.01 | $-0.049994$ | 0.01 | $-0.125889$ | 0.01 | $-0.050028$ | 0.01 | $-0.126000$ |
| | 0.1 | $-0.047948$ | 0.1 | $-0.115064$ | 0.1 | $-0.047802$ | 0.1 | $-0.121284$ |
| | 0.5 | $-0.135115$ | 0.5 | $-0.004699$ | 0.5 | $-0.265101$ | 0.5 | $-0.001122$ |
| | 1.0 | $-0.284395$ | 1.0 | $-0.002698$ | 1.0 | $-0.271693$ | 1.0 | $-0.005325$ |
| | 2.5 | $-0.271292$ | 2.5 | $-0.276510$ | 2.5 | $-0.269729$ | 2.5 | $-0.270111$ |
| $2p \to 3d$ | 0.01 | 1.062893 | 0.01 | 0.511654 | 0.01 | 1.062940 | 0.01 | 0.511243 |
| | 0.1 | 1.059233 | 0.1 | 0.563033 | 0.1 | 1.061126 | 0.1 | 0.525528 |
| | 0.5 | 1.039334 | 0.5 | 1.058267 | 0.5 | 1.033436 | 0.5 | 1.037782 |
| | 1.0 | 1.032612 | 1.0 | 1.035607 | 1.0 | 1.029282 | 1.0 | 1.028293 |
| | 2.5 | 1.030470 | 2.5 | 1.030594 | 2.5 | 1.030324 | 2.5 | 1.030294 |
| $3d \to 2p$ | 0.01 | $-0.637736$ | 0.01 | $-0.306992$ | 0.01 | $-0.637764$ | 0.01 | $-0.306745$ |
| | 0.1 | $-0.635539$ | 0.1 | $-0.337820$ | 0.1 | $-0.636675$ | 0.1 | $-0.315316$ |
| | 0.5 | $-0.623600$ | 0.5 | $-0.634960$ | 0.5 | $-0.620061$ | 0.5 | $-0.622669$ |
| | 1.0 | $-0.619567$ | 1.0 | $-0.621364$ | 1.0 | $-0.617569$ | 1.0 | $-0.616975$ |
| | 2.5 | $-0.618282$ | 2.5 | $-0.618356$ | 2.5 | $-0.618194$ | 2.5 | $-0.618176$ |
| $3d \to 4f$ | 0.01 | 1.358996 | 0.01 | 1.377113 | 0.01 | 1.358989 | 0.01 | 1.377140 |
| | 0.1 | 1.358827 | 0.1 | 1.374730 | 0.1 | 1.358891 | 0.1 | 1.376182 |
| | 0.176 | 1.352618 | 0.15 | 1.372349 | 0.12602 | 1.354588 | 0.157 | 1.372370 |
| | 0.1931 | 1.347244 | 0.255 | 1.364948 | 0.12638 | 1.354413 | 0.16724 | 1.369061 |
| | 0.19462 | 1.346487 | 0.29949 | 1.354689 | 0.12671 | 1.354225 | 0.16812 | 1.368517 |

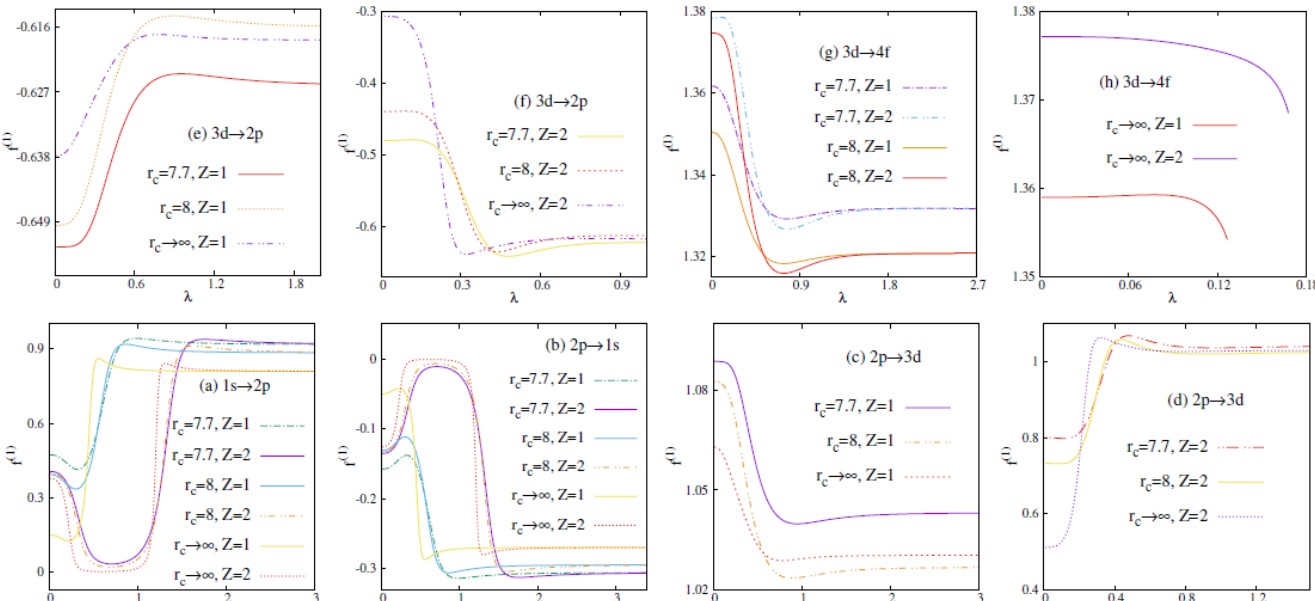

**Figure 4.** Plot of $f^{(1)}$ as function of $\lambda$ in ECSCPWS potential for selected transitions mentioned in the panels, for $Z = 1, 2$. See text for details.

### 3.3. Polarizability

In this subsection now, we proceed to discuss $\alpha^{(1)}$ for $1s, 2p, 2s$ states. It can have $(-)$ve as well as $(+)$ve values. When emission predominates over absorption, we derive

$(-)$ve values. On the contrary, a $(+)$ve values indicates that, absorption contribution has overcome the respective emission part. For the ground $1s$ state, it is always $(+)$ve. However, for excited states, it can have either sign. Here, results are presented for $1s, 2p, 2s$ states involving $r_c = 7.7, 8$, infinity having $Z$ values 1 and 2.

In Table 4, we have shown the results for $\alpha^{(1)}$ for these three states at $r_c = 7.7, 8$ and $Z = 1, 2$. Columns three to six represent DHPWS while ECSCPWS results are tabulated in columns seven to ten. At a given $Z$ and $r_c$, in $1s$ state it always increases with $\lambda$. At a given $r_c$, these decrease with $Z$. Conversely, at a given $Z$, it increases with $r_c$. Interestingly, for $2s$ and $2p$, the pattern behavior of $\alpha^{(1)}$ is not that straightforward. Depending upon the $\lambda$ values it can have either a $(-)$ve or $(+)$ve sign. However, the absolute numerical values abate with $Z$ and progress with $r_c$. Table 5 indicates $\alpha^{(1)}$ for both the plasmas in $1s, 2s, 2p$ states at $r_c \to \infty$ at $Z = 1, 2$. In $1s$ state, as usual it grows with a rise in $\lambda$. However, in the other two cases, we observe both $(+)$ve and $(-)$ve results.

To probe further, we have now plotted $\alpha^{(1)}$ as a function of $\lambda$ at $r_c = 7.7, 8$ and infinity, keeping $Z$ fixed at 1, 2. Figures 5 and 6 represent DHPWS and ECSCPWS cases, respectively. Panels (a,b) correspond to $1s$ state. Similarly, panels (c–e) give the plots for $2p$ state, and finally $2s$ results are produced in panels (f–i). It complements the conclusion discussed in Tables 4 and 5. However, for $2s$, and $2p$ curves, one can observe discontinuities. This may appear due to a sign change in $\alpha^{(1)}$.

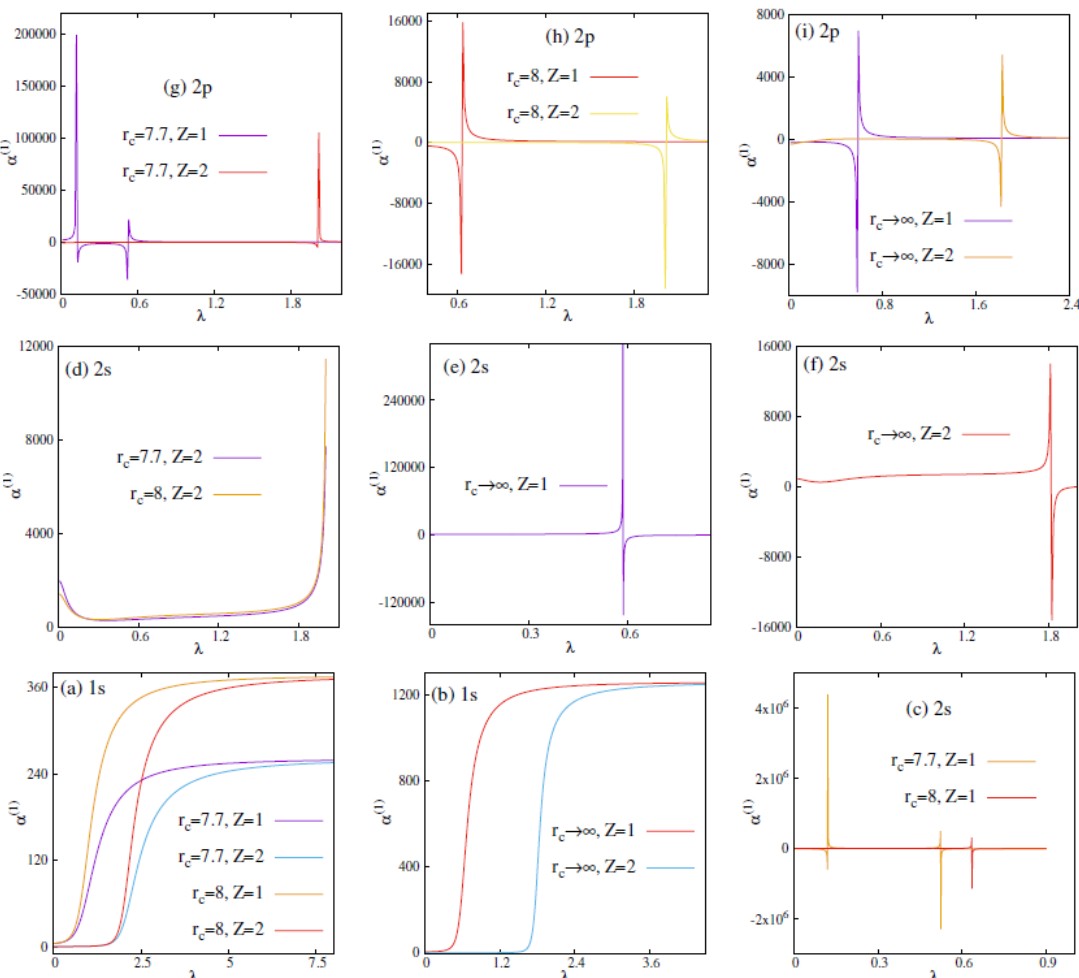

**Figure 5.** Plot of $\alpha^{(1)}$ as function of $\lambda$ in DHPWS potential, for $1s, 2s, 2p$ states as mentioned in the panels, for $Z = 1, 2$. See text for details.

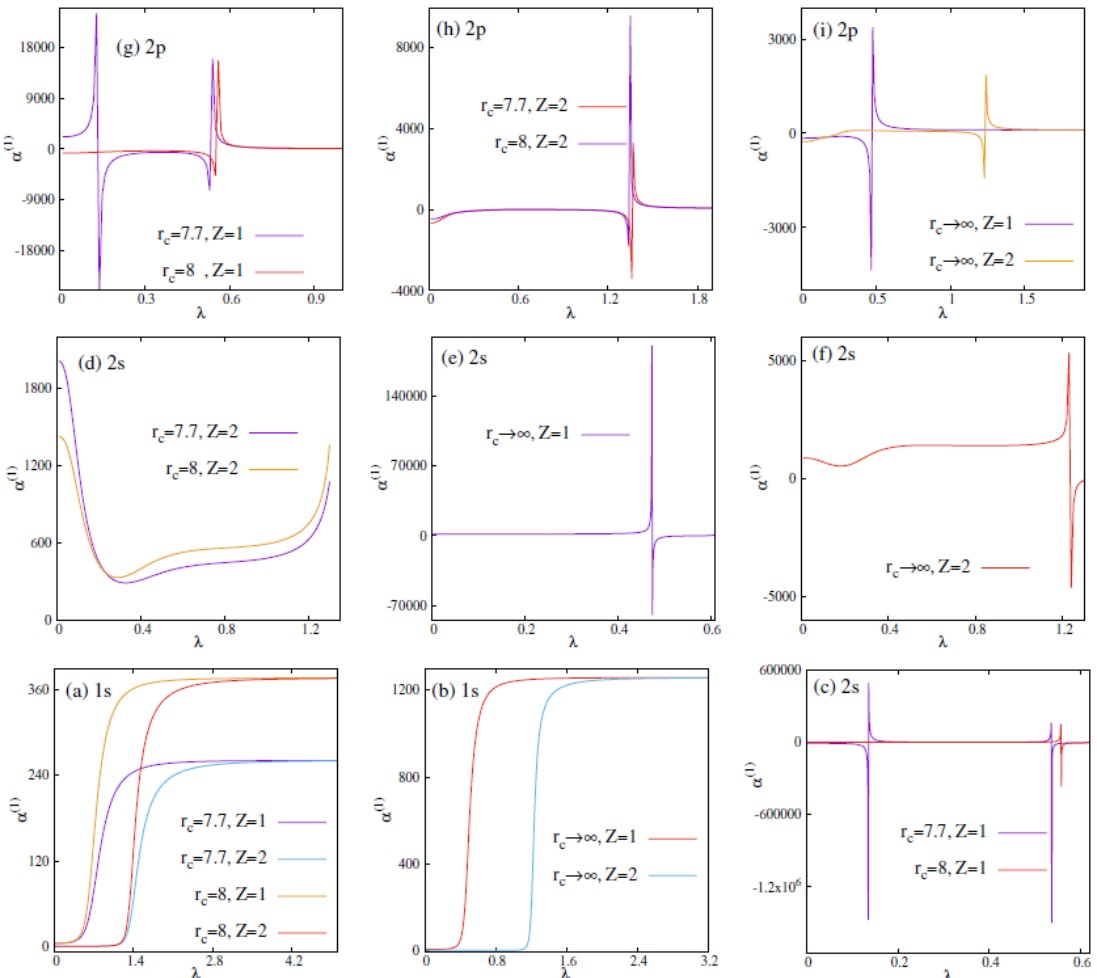

**Figure 6.** Plot of $\alpha^{(1)}$ as function of $\lambda$ in ECSCPWS potential, for $1s, 2s, 2p$ states as mentioned in the panels, for $Z = 1, 2$. See text for details.

**Table 4.** $\alpha^{(1)}$ values for DHPWS and ECSCPWS for $r_c = 7.7$ & 8 and $Z = 1, 2$, using $L = 1$ and $N = 300$.

| State | $\lambda$ | DHPWS | | | | ECSCPWS | | | |
|---|---|---|---|---|---|---|---|---|---|
| | | $r_c = 7.7$ | | $r_c = 8$ | | $r_c = 7.7$ | | $r_c = 8$ | |
| | | $Z = 1$ | $Z = 2$ | $Z = 1$ | $Z = 2$ | $Z = 1$ | $Z = 2$ | $Z = 1$ | $Z = 2$ |
| $1s$ | 0.01 | 4.657831 | 0.28128622 | 4.746857 | 0.2812862 | 4.6554220 | 0.2812521 | 4.7442022 | 0.2812521 |
| | 0.1 | 4.881931 | 0.28451405 | 4.994924 | 0.2845141 | 4.7251882 | 0.2817758 | 4.8232375 | 0.2817759 |
| | 0.5 | 12.2287 | 0.35794112 | 14.6516 | 0.3579433 | 17.04331 | 0.3321934 | 24.00875 | 0.3321963 |
| | 1.0 | 78.7749 | 0.69849914 | 140.048 | 0.6990358 | 202.0018 | 1.1380112 | 320.2554 | 1.1591198 |
| | 2.5 | 230.696 | 128.19895 | 345.626 | 226.2246 | 259.3304 | 251.0688 | 375.0790 | 367.0643 |
| | 3.0 | 240.868 | 187.73712 | 356.253 | 299.8536 | 260.2497 | 256.3160 | 375.9265 | 372.1057 |
| $2s$ | 0.01 | −6323.6041 | 1974.66567 | 2449.20791 | 1407.62883 | −6259.74897 | 2014.6251 | 2455.03251 | 1427.48523 |
| | 0.1 | −25,299.096 | 762.542243 | 2096.11187 | 671.915445 | −12,087.9810 | 1214.6507 | 2197.55093 | 975.180845 |
| | 0.5 | 18,546.933 | 297.184946 | 2569.87340 | 369.786944 | 4830.94856 | 368.81438 | 2570.99350 | 481.397967 |
| | 1.0 | −252.97113 | 421.447127 | −232.042760 | 527.012512 | −26.4762447 | 481.38339 | −9.29681306 | 592.581494 |
| | 2.5 | 9.5211800 | −128.377389 | 18.4890829 | −86.8752095 | 29.6816393 | 21.677927 | 36.7365367 | 29.6077279 |
| | 3.0 | 16.629850 | −29.6973425 | 24.7934626 | −13.1804006 | 30.8831256 | 27.068895 | 37.7878081 | 34.3689306 |
| $2p$ | 0.01 | 2191.410 | −643.03504 | −702.8776 | −452.1602 | 2170.076 | −656.3755 | −704.8910 | −458.8075 |
| | 0.1 | 8520.618 | −237.05905 | −579.3616 | −204.1988 | 4114.992 | −388.9010 | −615.7853 | −306.8603 |
| | 0.5 | −6061.053 | −24.508518 | −694.8934 | −13.59247 | −1473.332 | −1.579597 | −678.1659 | 9.116706 |
| | 1.0 | 204.2574 | −8.8332529 | 219.5863 | −0.670906 | 100.0618 | −6.121894 | 97.77017 | −1.240847 |
| | 2.5 | 79.78501 | 152.51948 | 81.67296 | 148.6793 | 67.74707 | 71.51052 | 70.12450 | 73.50398 |
| | 3.0 | 75.37053 | 102.96430 | 77.42353 | 103.0182 | 67.26899 | 69.03491 | 69.71802 | 71.31396 |

**Table 5.** $\alpha^{(1)}$ values for DHPWS and ECSCPWS for $r_c \rightarrow \infty$ and $Z = 1, 2$, using $L = 1$ and $N = 300$.

| State | DHPWS | | | | ECSCPWS | | | |
|---|---|---|---|---|---|---|---|---|
| | $\lambda$ | $Z = 1$ | $\lambda$ | $Z = 2$ | $\lambda$ | $Z = 1$ | $\lambda$ | $Z = 2$ |
| | 0.01 | 5.041120 | 0.01 | 0.2812863 | 0.01 | 5.0373245 | 0.01 | 0.2812522 |
| | 0.1 | 5.405572 | 0.1 | 0.2845141 | 0.1 | 5.1638458 | 0.1 | 0.2817760 |
| 1s | 0.5 | 71.4353 | 0.5 | 0.3579454 | 0.5 | 553.3792 | 0.5 | 0.3321994 |
| | 1.0 | 1076.49 | 1.0 | 0.7000504 | 1.0 | 1241.072 | 1.0 | 1.2523499 |
| | 2.5 | 1242.28 | 2.5 | 1183.049 | 2.5 | 1257.825 | 2.5 | 1254.207 |
| | 3.0 | 1247.93 | 3.0 | 1220.684 | 3.0 | 1258.140 | 3.0 | 1256.377 |
| | 0.01 | 1480.865 | 0.01 | 871.382822 | 0.01 | 1480.571 | 0.01 | 877.9676 |
| | 0.1 | 1507.894 | 0.1 | 578.058339 | 0.1 | 1497.317 | 0.05 | 845.6145 |
| 2s | 0.3 | 1720.036 | 0.5 | 1060.98797 | 0.2 | 1587.219 | 0.09 | 738.4854 |
| | 0.5 | 2721.953 | 1.0 | 1338.07651 | 0.3 | 1771.873 | 0.3 | 940.9011 |
| | 0.7 | $-326.1830$ | 1.5 | 1503.41076 | 0.5 | $-1276.650$ | 0.7 | 1404.159 |
| | 0.9 | 23008.31 | 2.0 | $-23.222516$ | 0.6 | 259.9035 | 1.3 | $-65.30549$ |
| | 0.01 | $-150.7481$ | 0.01 | $-262.0658419$ | 0.01 | $-150.9188$ | 0.01 | $-264.3678$ |
| | 0.1 | $-139.2984$ | 0.1 | $-152.6892619$ | 0.1 | $-140.9449$ | 0.1 | $-201.3010$ |
| 2p | 0.5 | $-443.5363$ | 0.5 | 69.93496507 | 0.5 | 786.5248 | 0.5 | 87.43362 |
| | 1.0 | 168.1365 | 1.0 | 64.90555224 | 1.0 | 114.3632 | 1.0 | 55.87104 |
| | 2.5 | 111.5520 | 2.5 | 133.2633406 | 2.5 | 106.2537 | 2.5 | 107.5945 |
| | 3.0 | 109.6328 | 3.0 | 119.4094711 | 3.0 | 106.1153 | 3.0 | 106.7672 |

## 4. Conclusions

Multi-pole oscillator strength and polarizability are estimated for H-like plasmas inside a fullerene cavity under high pressure and with special emphasis on excited states. The desired confinement condition has been imposed primarily by trapping the system under a endohedral cavity. Illustrative calculations have been undertaken for $1s, 2s, 2p, 3d$ states. Negative values of $f^{(1)}$ and $\alpha^{(1)}$ are reported here for certain excited states. The qualitative behavior of $\alpha_{1s}^{(1)}$ does not change with $r_c$ or $Z$. Moreover, $\lambda^{(c)}$ has been studied in a free condition using $S_r$. A thorough analysis reveals that, under a fullerene cavity $1s, 2p, 3d$ states will never be deleted. However, for other states, at $\lambda^{(c)}$, $S_r$ jumps suddenly indicating a first-order phase transition. A detailed inspection of $f^{(k)}$ and $\alpha^{(k)}$ in many-electron systems under confinement would throw some light on their understanding. Investigation of several other properties like the photo-ionization cross-section, the mean excitation energy, and the electronic stopping cross-section may also be undertaken in future.

**Author Contributions:** Conceptualization, N.M.; Formal analysis, N.M.; Investigation, N.M.; Resources, A.K.R.; Data curation, S.C.; Writing–original draft, N.M.; Writing—review and editing, S.C. and A.K.R.; Supervision, A.K.R.; Funding acquisition, A.K.R. All authors have read and agreed to the published version of the manuscript.

**Funding:** This research was funded by SERB-DST, India (sanction order: CRG/2019/000293) and CSIR, New Delhi (grant number: 01(3027)/21/EMR-II).

**Data Availability Statement:** Data will be available on request.

**Acknowledgments:** The authors thank two anonymous referees for their constructive comments.

**Conflicts of Interest:** The authors declare no conflict of interest.

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
