# Peer review of "Hydrogen-like Plasmas under Endohedral Cavity"

_quantumrep, doi:10.3390/quantum5020030_

Round 1

Reviewer 1 Report

Referee Report

Mss Entitled: Hydrogen-like plasmas under endohedral cavity

Authors          :Saptarshi Chowdhury, Neetik Mukherjeeand Amlan K. Roy

This manuscript contains interesting new results on the endohedrally confined H-atom under the dilute and dense plasma environments. The authors are encouraged to consider the following suggestions/remarks:

(1)  On all the table headings,  the GPS mapping parameter L and the number of grid points, N,  used should be stated.

(2)  Comment how was the value of rc  (a.u.) = 7.7 and 8 generated in the actual computations. Which particular L & N was  employed ?

(3)  In the celebrated early work of C. Stubbins, Phys. Rev. 48, 220 (1993) with reference to the Eq. 3.9 one finds that the inequality is obeyed , e.g. in N. Mukherjee and A. Roy Phys. Rev. A 104, 042803 (2021) , Table I for the entries under  2c, 3f, and 4j . However, in presence of the attractive shell potential (WS) in the manuscript under review, the entries in Table I do not obey the comparison theorem  arguments highlighted by Stubbins. Authors may consider explaining this departure in the specific cases under Table I . Note that for Z=1 & 2 for 2s and Z=1 for 3s the energy ordering is  E(DHPWS) < E(ECSPWS) thereafter the order is found to be either <  or   > .

(4)  A reference to  this work may be included: Alkali atoms confined to a sphere and to a fullerene cage , S.H. Patil , K.D. Sen and Y.P. Varshni , Can. J. Phys. 83 , 919 (2005 ).

I suggest minor revision.

Author Response

\documentclass[12pt]{article}
\usepackage{xcolor}
\usepackage{amsmath}
\newcommand{\rvec}{\mathrm {\mathbf {r}}} 
\newcommand{\pvec}{\mathrm {\mathbf {p}}} 
%%% please use this above. 

\def\ul{\underline}
\textwidth 7.0in
\textheight 9.5in
\hoffset -1in
\voffset -1in 

\begin{document}
%%%%%%%%%%%%%%%%%%%%%%%%%%%%%%%%%%%%%%%%%%%%%%%%%%%%%%%%

\begin{center}
{\underline {\bf {Response to Referee Report: quantumrep-2295719}}}
\end{center}

First of all, it is a pleasure to thank both the Referees for their valuable, detailed comments.
All the changes are marked with blue color. 

\begin{enumerate}
\item
\underline{Response to Referee 1}: 

\begin{enumerate}
\item
Referee says \\
``On all the table ..... be stated" 

Reply: \\
Numerical calculations are performed by choosing $L=1$ and $N=300$. It has now 
been mentioned in the revised manuscript.  

\item
Referee says \\
``Comment ............employed?". 

Reply: \\
Experimental data suggests that, in a fullerene system the sum of cage radius and thickness of the shell is $\approx$ 7.7 a.u. Furthermore, the boundary has been slightly relaxed
to 8 a.u. to record the effect of $r_{c}$. Additionally, calculation is also done for $r_{c} \rightarrow \infty$. It would be noted that, the former case represent a hard 
confinement. However, latter explains a soft confinement.       

\item
Referee says \\
``In the celebrated ............ to be either $<$ or $>$."

Reply: \\
The theorem discussed in Phys. Rev. A 48, 220 (1993), is valid when both the potentials have same $\lambda$ value. In Table~I, critical screening constant and the respective 
energy at that point are reported, which differs from state to state. For a given state, of course, $\lambda_{c}$ are not same for DHPWS and ECSPWS. Additionally we have performed some tests for a fixed $\lambda$ to check the validity of the bound in DHPWS and ECSPWS. The bound 
was satisfied in all cases.  

\item
Referee says \\
``A reference ........ 919 (2005)."

Reply: \\
It has been added in present manuscript (Ref. [17]).

\end{enumerate}

\item
\underline{Response to Referee 2}: 

\begin{enumerate}
\item
Referee says, \\
``Line 85.............values?" 

Reply: \\
This first question has already been discussed in the second point of the first referee's comment. 

The logic behind the choice of $r_c =7.7$, 8 and $5 \times 10^5$ has already been mentioned in 1(b).  The latter represents the 
corresponding free systems.

A change in $r_c$ from above would change the numerical results. These were chosen as representative cases. Our motivation was 
to study the variation of $\lambda$ keeping $r_c$ fixed at those selected values. 

\item
Referee says, \\
``Line 115..........potential?." 

Reply: \\
The parameters we used in this communication suffice for historical use and experimental fitting. Although the thickness of the endohedral shell $\Delta$ and the radius of the endohedral shell 
$R_c$ are fixed and cannot be changed as they are directly incorporated from experimental data, one can obviously tweak the well-depth parameter $V_0$. In this case, we have considered a 
particular value for $V_0$ because it was tested repeatedly and seems to be a good fitting for the experimental data. As for the smoothing parameter $\gamma$, we have kept it 0.1, because for less 
than this, the attractive shell becomes close to a square well and for more than or close to 0.2, it behaves like a Gaussian well.

From reference [71]-[76] it can be mentioned that, Wood-Saxon potential is a better model compared to the model discussed in reference [73]. This has been discussed in the second paragraph of 
introduction section. 

\item
Referee says, \\
``Line 120..........what is (i)?." 

Reply: \\
It has been corrected.  

\item
Referee says, \\
``Line 138..........paper." 

Reply: \\
$\lambda^{(c)}$ is now defined in Sec.~I, page 3. 

\item
Referee says, \\
``Equation (14)...... the paper."

Reply: \\
It is true that results are given only for $S_r$. However, for the sake of completeness, we believe it is alright to keep the entire equation. 

\item
Referee says, \\
``Line 153..........and (+)ve." 

Reply: \\
It was pointed that, bound states have $-$ve energy and continuum state possesses $+$ve energy. However, in line 234, discussion 
has been made for dipole polarizability. For a given state it can have either $+$ve or $-$ve value. 

\item
Referee says, \\
``Line 175..........here?." 

Reply: \\
$\ell= 0, 1, 2$ correspond to states having azimuthal quantum number values as 0, 1, 2 respectively. 

\item
Referee says, \\
``Line 182..........2018, 3, 227." 

Reply: \\
The mentioned footnotes refer to those in Table~I, not of the reference [62].        

\item
Referee says, \\
``Line 239..........here." 

Reply: \\
It is done.          

\end{enumerate}
\end{enumerate}

\end{document}

Author Response

(The authors gave the same response as above.)

Round 2

Reviewer 1 Report

The authors have satisfactorily addressed the queries raised earlier.

I recommend publication. 

Reviewer 2 Report

The changes of the authors have addressed all of the issues in my initial report.

This paper is ready for publication.